# A Comparison of Hypofractionated and Twice-Daily Thoracic Irradiation in Limited-Stage Small-Cell Lung Cancer: An Overlap-Weighted Analysis

**DOI:** 10.3390/cancers13122895

**Published:** 2021-06-09

**Authors:** Michael Yan, Samantha Sigurdson, Noah Greifer, Thomas A. C. Kennedy, Tzen S. Toh, Patricia E. Lindsay, Jessica Weiss, Katrina Hueniken, Christy Yeung, Vijithan Sugumar, Alexander Sun, Andrea Bezjak, B. C. John Cho, Srinivas Raman, Andrew J. Hope, Meredith E. Giuliani, Elizabeth A. Stuart, Timothy Owen, Allison Ashworth, Andrew Robinson, Fabio Ynoe de Moraes, Geoffrey Liu, Benjamin H. Lok

**Affiliations:** 1Department of Oncology, Cancer Centre of Southeastern Ontario, Queen’s University, Kingston, ON K7L 5P9, Canada; michael.yan@kingstonhsc.ca (M.Y.); samantha.sigurdson@kingstonhsc.ca (S.S.); thomas.kennedy@kingstonhsc.ca (T.A.C.K.); timothy.owen@kingstonhsc.ca (T.O.); allison.ashworth@kingstonhsc.ca (A.A.); andrew.robinson@kingstonhsc.ca (A.R.); fabio.ynoedemoraes@kingstonhsc.ca (F.Y.d.M.); 2Department of Mental Health, Bloomberg School of Public Health, Johns Hopkins University, Baltimore, MD 21205, USA; ngreife1@jhu.edu (N.G.); estuart@jhu.edu (E.A.S.); 3The Medical School, University of Sheffield, Sheffield S10 2RX, UK; tzen.szen@gmail.com; 4Division of Medical Oncology and Hematology, Princess Margaret Cancer Centre, Toronto, ON M5G 2M9, Canada; Katrina.hueniken@uhnresearch.ca (K.H.); Geoffrey.Liu@uhn.ca (G.L.); 5Radiation Medicine Program, Princess Margaret Cancer Centre, Toronto, ON M5G 2M9, Canada; Patricia.lindsay@rmp.uhn.ca (P.E.L.); Alex.Sun@rmp.uhn.ca (A.S.); andrea.bezjak@rmp.uhn.ca (A.B.); john.cho@rmp.uhn.ca (B.C.J.C.); srinivas.raman@rmp.uhn.ca (S.R.); andrew.hope@rmp.uhn.ca (A.J.H.); Meredith.Giuliani@rmp.uhn.ca (M.E.G.); 6Department of Radiation Oncology, University of Toronto, Toronto, ON M5T 1P5, Canada; 7Department of Biostatistics, Princess Margaret Cancer Centre, Toronto, ON M5G 2M9, Canada; jessica.weiss@uhnresearch.ca; 8Department of Laboratory Medicine and Pathobiology, University of Toronto, Toronto, ON M5G 0A4, Canada; christy.yeung@mail.utoronto.ca; 9Department of Physiology and Pharmacology, Western University, London, ON N6A 5C1, Canada; vsuguma@uwo.ca; 10Institute of Medical Science, Faculty of Medicine, University of Toronto, Toronto, ON M5S 1A8, Canada; 11Department of Medical Biophysics, University of Toronto, Toronto, ON M5G 1L7, Canada

**Keywords:** small-cell lung cancer, hypofractionation, radiotherapy, propensity score

## Abstract

**Simple Summary:**

The optimal thoracic radiotherapy schedule for limited-stage small cell lung cancer (LS-SCLC) patients remains controversial. We conducted a propensity score adjusted analysis of LS-SCLC patients treated at our institutions with 40Gy/15 fractions versus 45Gy/30 twice daily. After overlap weighting for clinical and treatment variables and attaining good balance, we did not find a significant difference in overall survival, locoregional recurrence risk, thoracic response, or ≥grade 3 toxicity. Moderate hypofractionation, with its similar outcomes and logistical advantages, may present a reasonable alternative to twice daily radiotherapy.

**Abstract:**

Despite evidence for the superiority of twice-daily (BID) radiotherapy schedules, their utilization in practice remains logistically challenging. Hypofractionation (HFRT) is a commonly implemented alternative. We aim to compare the outcomes and toxicities in limited-stage small-cell lung cancer (LS-SCLC) patients treated with hypofractionated versus BID schedules. A bi-institutional retrospective cohort review was conducted of LS-SCLC patients treated with BID (45 Gy/30 fractions) or HFRT (40 Gy/15 fractions) schedules from 2007 to 2019. Overlap weighting using propensity scores was performed to balance observed covariates between the two radiotherapy schedule groups. Effect estimates of radiotherapy schedule on overall survival (OS), locoregional recurrence (LRR) risk, thoracic response, any ≥grade 3 (including lung, and esophageal) toxicity were determined using multivariable regression modelling. A total of 173 patients were included in the overlap-weighted analysis, with 110 patients having received BID treatment, and 63 treated by HFRT. The median follow-up was 20.4 months. Multivariable regression modelling did not reveal any significant differences in OS (hazard ratio [HR] 1.67, *p* = 0.38), LRR risk (HR 1.48, *p* = 0.38), thoracic response (odds ratio [OR] 0.23, *p* = 0.21), any ≥grade 3+ toxicity (OR 1.67, *p* = 0.33), ≥grade 3 pneumonitis (OR 1.14, *p* = 0.84), or ≥grade 3 esophagitis (OR 1.41, *p* = 0.62). HFRT, in comparison to BID radiotherapy schedules, does not appear to result in significantly different survival, locoregional control, or toxicity outcomes.

## 1. Introduction

Small-cell lung cancer is an aggressive histology of lung cancer with poor prognosis. Only about a third of patients have limited-stage disease (LS-SCLC) at diagnosis, with the potential to pursue curative intent treatment. Concurrent chemotherapy and radiation (CRT) remain a standard therapy in the management of patients with limited-stage small-cell lung cancer [1]. The addition of thoracic irradiation to chemotherapy has been shown to improve survival in this patient population [2].

Several trials have investigated the optimal radiotherapy schedule. Turrisi et al. compared hyperfractionated, twice-daily radiotherapy (BID) with a conventionally fractionated once-daily schedule, both to 45 Gy. They established superior survival and disease-free survival with the former schedule, albeit with a higher rate of toxicities [3]. Subsequently, the CONVERT trial did not demonstrate superior survival with dose escalation of daily radiotherapy to 66 Gy when compared to the 45 Gy/30 BID schedule [4].

Despite these outcomes, adoption of BID regimens into clinical practice has been limited. Barriers include the logistical complexities from both a provider and patient perspective, as well as the potential for increased toxicity. Recent survey studies from Canada and the US report that only about a quarter of physicians routinely utilize BID schedules [5,6].

In contrast, hypofractionated radiotherapy (HFRT) schedules are frequently utilized. A small, phase 2 randomized trial comparing BID and HFRT did not show a significant survival difference between the two cohorts [7]. Further observational studies have supported the effectiveness of HFRT in comparison to BID [8], or conventional fractionation schedules in concurrent chemoradiotherapy [9].

One common HFRT schedule is 40 Gy in 15 daily fractions, initially established in a trial by the Canadian Cancer Trials Group (CCTG) [10]. This schedule is frequently used at our institutions, as well as nationally within Canada; despite the lack of comparative prospective evidence [5]. The objective of the current study is to compare the outcomes of LS-SCLC patients treated with HFRT versus BID schedules of CRT at two Canadian institutions, while employing the use of propensity score methods to better adjust for confounding variables. 

## 2. Materials and Methods

### 2.1. Data Sources

We reviewed institutional databases from Princess Margaret Hospital (PMH) and Kingston Health Sciences Centre (KHSC) for patients with LS-SCLC who were treated with curative intent chemoradiation between January 2007 and November 2019. HFRT was routinely delivered at both institutions until 2006, in which, thereafter, BID began to be more regularly adopted into institutional practice. Generally, most suitable patients who were young and fit were offered BID treatment. Eligible patients may have declined due to logistical reasons, and opted for HFRT instead. A subset of patients received surgical resection upfront for solitary parenchymal disease or had been presumed to have non-SCLC; these patients received adjuvant chemoradiotherapy. Thoracic radiotherapy techniques varied from conventional to intensity-modulated radiotherapy (IMRT) and dependent on the era of treatment. All data were retrospectively acquired and managed using REDCAP electronic data capture tools [11]. This study was approved by the research ethics boards of both institutions.

Patients were excluded if they had extensive-stage disease, treated with palliative intent, or did not receive both chemotherapy and radiation. Furthermore, we only included patients treated with radiotherapy schedules of either 45 Gy in 30 twice-daily fractions (BID) or 40 Gy in 15 once-daily fractions (HFRT). Clinical, treatment, and outcomes details were retrospectively collected for all patients. 

Clinical staging was determined using the 8th edition of the American Joint Committee on Cancer (AJCC) staging system [12]. Comorbidities were calculated using a modified Charlson Comorbidity Index (mCCI) score [13]. Radiotherapy was considered concurrent if the course was delivered prior to the final cycle of chemotherapy. Start of any treatment to end of radiotherapy (SER) was defined as the time in days from treatment start to the last thoracic radiotherapy fraction [14].

### 2.2. Data Processing 

Missing covariate data were imputed using multiple imputation by chained equations, with all variables in the analysis used in the imputation procedure. Imputations were generated using random forest classification, and 200 imputations were generated. The fraction of missing data was <5% for all variables, which is considered low [15]. 

### 2.3. Propensity Score Methods and Diagnostics 

Various propensity score models were generated with radiotherapy schedule (BID vs. HFRT) as the treatment indicator (dependent variable). Age, gender, surgery utilization, stage, smoking status, smoking pack years, ECOG status, CCI score, pretreatment brain imaging modality, cycles of chemotherapy, concurrent versus sequential chemoradiation, SER, positron emission topography (PET) utilization, radiotherapy technique, image-guided radiotherapy (IGRT) utilization, and 4D-CT utilization were used as covariates (independent variables) in model generation. PCI utilization was not included in model generation, as the determination of offering PCI is dependent upon response to chemoradiation, and therefore is not independent of the effects of the radiotherapy schedule treatment variable. PCI utilization was, however, adjusted for in subsequent multivariable regression modelling, and independently analyzed for association with OS [16]. Both matching and weighting methods were assessed, with the approach resulting in best covariate balance selected. A standardized mean difference (SMD) <0.1 is considered to indicate reasonably good balance [17].

Propensity score weights estimated as overlap weights were selected to be used in subsequent analyses as they resulted in the best covariate balance and effective sample size among the several attempted adjustment methods. Overlap weights allow us to estimate the average treatment effect in the overlap population (ATO), which corresponds to those patients approximately equally likely to receive either treatment (i.e., at clinical equipoise) [18,19]. Each patient’s weight is proportional to the probability of that patient being assigned to the opposite radiotherapy group, so that patients who are less likely to be assigned to their actual group hold a greater weight than a patient who was more likely. For example, a patient treated in a more contemporary year (who would be more likely to have received BID treatment) would hold less weight if they had in actuality received BID treatment, and more weight if they were treated by HFRT. 

Overlap weighting was performed individually within each imputed dataset. Effect estimates of the outcomes, described in the following section, were then obtained within each imputed dataset and combined across datasets using the multiple imputation combining rules. 

### 2.4. Outcome Analysis 

Descriptive statistics were generated for the unweighted cohort. The Mann–Whitney U test and Fisher’s exact test were used to compare continuous and categorical covariates between the BID and HFRT cohorts, respectively. 

The primary outcome of interest was overall survival (OS). Secondary outcomes included locoregional recurrence (LRR) risk, thoracic response to chemoradiation, any grade 3 or greater toxicity, grade 3 or greater lung toxicity (LT), and grade 3 or greater esophageal toxicity (ET). Both unweighted and overlap-weighted analyses were performed for each endpoint. 

OS curves were estimated using the Kaplan–Meier method, defined from the date of histologic diagnosis to the date of last follow-up or death from any cause. Differences in survival curves were evaluated using the adjusted log-rank test [20]. LRR risk was estimated from cumulative incidence functions based on sub-distribution hazards. LRR was defined from date of diagnosis to date of local or regional failure, whichever happened first; death without locoregional relapse was considered a competing event. Thoracic response was determined from radiological reporting and based on the RECIST 1.1 criteria [21]. Response was stratified into good response, consisting of complete or partial response (CR or PR), and poor response, consisting of stable or progressive disease (SD/PD). Toxicities were graded using the Common Terminology Criteria for Adverse Events (CTCAE) version 5 [22]. Lung toxicity and esophageal toxicity in the form of radiation pneumonitis and esophagitis, respectively, were assessed. 

Univariable and multivariable regression analyses using the unweighted and overlap-weighted cohorts were modelled for all endpoints. Covariates judged to be clinically associated with each outcome were determined a priori and included in the models. Cox proportional hazards regression was used to generate a model for OS. Fine–Gray competing risk regression was used to determine the effect estimate of the treatment variable on LRR risk [23]. Thoracic response to chemoradiation as well as ≥grade 3 toxicities, including LT and ET events, were modelled using logistic regression. Robust standard errors were used to account for the weights. 

### 2.5. Sensitivity to Unobserved Confounding

Sensitivity to unobserved confounding was assessed using e-values. The e-value represents the minimum magnitude of association, on a risk ratio scale, that a confounding variable not included in our models (unobserved confounding) must have with the treatment and outcome to alter our conclusions [24]. 

All statistical analyses were conducted in R version 4.0.1 (R Foundation for Statistical Computing, Vienna, Austria). Multiple imputation was performed using the *mice* package, with estimated effects combined using the *mitools* (version 2.4) package [25,26]. Weighting was performed using the *MatchThem* (version 0.9.3) package [27]. Covariate balance was assessed using the *cobalt* (version 4.2.4) package. Survival analyses were performed using the *survival* (version 3.2.7) package [28]. E-values were assessed using the *EValue* (version 4.1.1) package [24].

## 3. Results

### 3.1. Patient Characteristics

A total of 229 patients with LS-SCLC were initially identified from the two institutional databases consecutively treated with curative intent between January 2007 and November 2019. After exclusions, 173 patients were eligible for overlap-weighting analysis, with 110 patients treated with BID fractionation and 63 treated with HFRT (Figure 1). The median follow-up for the entire cohort was 20.4 months from diagnosis to censoring. 

### 3.2. Overlap Weighting 

Baseline characteristics for the cohort as well as their balance are summarized in Table 1. In unweighted analyses, patients receiving BID radiotherapy were more likely to have PET scans, later year of diagnosis, concurrent chemotherapy, IGRT, IMRT, and shorter SER. Of note, PCI use was not significantly different between the two cohorts, with 73% and 65% of patients having received PCI in the BID and HFRT cohorts, respectively (*p* = 0.26). 

After overlap weighting, all covariates achieved exact mean balance with SMDs of 0. Minimal residual imbalances (all SMDs < 0.05) were observed in the square and cube of cycles of chemotherapy, and the cube of paraneoplastic syndrome and age (Appendix A) [19]. 

### 3.3. Overall Survival 

There was no significant difference in OS between patients in the BID and HFRT treatment groups (*p* = 0.93) in the unweighted analysis. The 5-year OS was 27.0% (95% CI, 20.1–36.3), 25.5% (95% CI, 17.1–37.9), and 29.3% (95% CI, 18.5–46.3%) for all, BID, and HFRT patients, respectively (Figure 2A). 

After overlap weighting, the 5-year OS was 24.3% (95% CI, 16.1–36.6) for all patients, and 22.1% (95% CI, 12.7–38.5) and 26.6% (95% CI, 14.4–49.0%) for BID and HFRT cohorts, respectively. Again, there was no significant differences between the OS curves of the two cohorts (*p* = 0.93) (Figure 2B). 

When stratifying by PCI use, there was a significant difference in OS between patients who received PCI and those who did not in the HFRT cohort (*p* = 0.004). This was not observed in the BID cohort (*p* = 0.4). However, after overlap weighting, there was no significant difference observed in either cohort (BID *p* = 0.09; HFRT *p* = 0.61) (Appendix A). 

The proportional hazards assumption was not violated for any included covariates in the regression modelling of OS. Univariable and multivariable cox regression was performed adjusting for age, PCI utilization, ECOG status, cycles of chemotherapy, stage, SER, concurrent or sequential chemoradiation, and mCCI score. The results did not reveal any significant differences in OS between BID- and HFRT-treated patients in both unweighted and weighted analyses (Table 2).

### 3.4. Locoregional Recurrence Risk 

As in OS, there was no significant difference in LRR risk between BID and HFRT cohorts in the unweighted (*p* = 0.96) and overlap-weighted (*p* = 0.40) analysis. In the unweighted analysis, the 5-year LRR risk was 63.6% (95% CI, 55.8–72.5) for all patients, 61.0% (95% CI, 51.4–72.5) for BID, and 68.7% (95% CI, 56.6–83.3) for the HFRT cohort (Figure 2C). In the weighted analysis, the risk was 68.9% (95% CI, 59.2–80.1), 68.9% (95% CI, 56.6–83.8), and 69.2% (95% CI, 55.3–86.6) for the same groups, respectively (Figure 2D). 

Univariable and multivariable Fine–Gray competing risk regression was performed for unweighted and weighted analyses, adjusting for stage, cycles of chemotherapy, concurrent versus sequential chemotherapy, SER, and radiotherapy technique. There were no significant differences detected between the two radiotherapy cohorts (Table 2). 

### 3.5. Thoracic Response to Chemoradiotherapy

A total of 161 patients (93%) had a good response to CRT, with 106 (96.4%) having received BID fractionation and 55 (87.3%) receiving HFRT in the unweighted analysis. There was a significantly higher proportion of good response observed in the BID compared to HFRT cohorts in the unweighted, univariable analysis (*p* = 0.03). However, after overlap weighting, this difference was no longer observed.

Similarly, after multivariable logistic regression analysis, adjusting for stage, cycles of chemotherapy, SER, concurrent versus sequential chemoradiation, and radiotherapy technique, there was no significant association between radiotherapy schedule and thoracic response, in both the unweighted and overlap-weighted analyses (Table 2). 

### 3.6. Toxicity

In the unweighted cohort, any grade 3 or greater toxicity was experienced by 34 (19.7%) patients, with 20 (18%) in the BID cohort and 14 (22%) in the HFRT cohort. There was no significant difference in the incidence between the two cohorts (*p* = 0.52). One patient in the BID cohort experienced both ≥grade 3 esophagitis and pneumonitis. The total incidence of ≥grade 3 esophagitis occurred in 23 (13%) patients, of which 14 (12.7%) and 9 (14.3%) were in the BID and HFRT cohorts, respectively. Again, there was no significant difference between the two groups (*p* = 0.77). Grade 3 or greater pneumonitis occurred in 12 (7%) patients, of which 7 (6%) were in the BID group and 5 (8%) were in the HFRT group, with no significant difference (*p* = 0.70). 

In both univariable and multivariable logistic regression, adjusting for stage, cycles of chemotherapy, IGRT use, concurrent versus sequential chemotherapy, and radiotherapy technique, there was no significant association between BID versus HFRT and any ≥grade 3 toxicity, ≥grade 3 esophagitis, or ≥grade 3 pneumonitis. This was consistent between unweighted and overlap-weighted analyses (Table 3).

### 3.7. Sensitivity to Unobserved Confounding 

The E-value estimates for the effect of radiotherapy schedule on OS and LRR as determined from multivariable regression models in the overlap-weighted cohort were 1.91 and 1.95, respectively (Appendix A). This suggests that only unobserved confounders with an association of nearly twofold, to both a patient’s propensity for receiving a certain radiotherapy schedule (HFRT or BID) as well as the outcomes of OS and LRR, would be able to abrogate the observed HRs for each endpoint. Weaker confounding variables would not be able to do so. Our E-value estimates suggest that our overlap-weighted propensity score analysis is robust. 

## 4. Discussion

Concurrent thoracic CRT is a critical component in the management of patients with LS-SCLC receiving curative intent treatment. The benefits of adding radiotherapy were established early on by a seminal meta-analysis of 13 randomized trials consisting of 2140 patients. Pignon et al. showed that, at 3 years, chemoradiation improved overall survival by 5.4% compared to chemotherapy alone [2]. Since then, trial efforts have shifted towards evaluating the details of chemoradiation; namely, the relative timing of the two modalities and the optimal dose fractionation. 

### 4.1. Establishment of BID as a Standard

Due to the rapidly proliferating nature of the disease, it was hypothesized that accelerated schedules may result in improved outcomes. Turrisi et al. established the superiority of BID treatment in 1.5 Gy fractions over 3 weeks in a randomized trial compared to daily conventionally fractionated treatment with 1.8 Gy fractions over 5 weeks. Both arms were treated to 45 Gy. The BID arm had superior survival, with a median OS of 23 months versus 19 months in the BID and daily arms, respectively (*p* = 0.04), albeit with higher rates of esophagitis [3]. One criticism of these results was the relatively lower biological equivalent dose (BED) given in the daily treatment arm as compared to BID treatment being delivered in a hyper-accelerated manner (i.e., more than 5 fractions per week) with greater BED. The CONVERT trial addressed this question by comparing the 45 Gy BID schema against a higher dose of 66 Gy in daily fractions. However, this trial did not show that daily fractionation was superior to BID fractionation. There was no significant difference in OS between the two arms (*p* = 0.14), although the median survival was nominally higher in the BID arm than the daily arm at 30 months versus 25 months, respectively [4]. Recently, a phase 2 randomized trial suggested a role for further dose escalation of twice-daily treatment. Grønberg et al. compared 60 Gy in 40 fractions BID with the conventional 45 Gy in 30 fractions BID, and found a survival difference at 2 years of 74.2% versus 48.1%, respectively (OR 3.09, *p* = 0.0005) [29].

### 4.2. Patterns of Practice

Despite high-level evidence, the BID regimen is not well adopted. A survey of Canadian radiation oncologists revealed that only about 30% of oncologists routinely prescribe 45 Gy/30 fractions BID for LS-SCLC patients, with the majority preferring 40 Gy/15 fractions [5]. Similarly, a more recent US survey found that more than three quarters of respondents prescribed daily treatment more commonly in their practice. The most common reasons for this were patient convenience, tolerability, and logistical simplicity. This survey also found that BID schedules were more frequently used in academic institutions [6]. This is corroborated by a National Cancer Database (NCDB) study in which BID utilization was found to be more likely in academic institutions (Odds Ratio [OR] 2.29, *p* < 0.001). In total, only 11% of eligible patients received BID schedules [30]. 

### 4.3. Evidence for HFRT 

One of the earliest prospective trials investigating radiotherapy timing in relation to chemotherapy utilized HFRT with 40 Gy/15 daily fractions; this provided prospective data with this fractionation schema [10]. From a biological stand point, the BED_10_ of 40 Gy/15 (50.67 Gy) is similar to that of 45 Gy/30 BID (51.75 Gy), and therefore creates a plausible basis for comparability. 

Subsequently, several retrospective studies support the use of HFRT thoracic radiotherapy schedules [8,31]. To date, only one prospective study, a phase 2 randomized control trial, compared BID and HFRT schedules head-to-head. Grønberg et al. randomized 157 patients, 84 patients to HFRT and 73 patients to BID, to be given concurrent with platinum-based chemotherapy. The study reported no significant difference in survival, with a median OS of 25 and 19 months in the BID and HFRT arms, respectively (*p* = 0.61). Similarly, no difference was observed in PFS or grade 3 toxicities. Patients treated with BID fractionation did, however, have high rates of complete response (33% vs. 13%, *p* = 0.003). This is consistent with our observations in the unweighted, univariable analysis; however, this relationship was no longer observed after overlap weighting or multivariable adjustment. The authors conclude that no firm inferences could be drawn from the study, and that a larger phase 3 trial is needed [7]. 

Like others, the current study did not observe any significant difference in clinical outcomes between HFRT and BID schedules. The median OS was around 2 years for both cohorts, similar to literature values [7,8,9,10,32,33,34] (Table 4). Likewise, there was no difference in LRR risk between the two fractionation schemes. This is supported by a modeling study by Li et al., who found no difference in tumor control probability (TCP) in the form of 2-year freedom from locoregional progression, between BID (34%) and HFRT (28%) arms (*p* = 0.44) [35]. Our competing risk regression models predicted risks of 27% and 25% for the same fractionation cohorts, and similarly, did not observe a significant difference between the two (*p* = 0.57). The rate of ≥grade 3 adverse events, in particular esophagitis and pneumonitis, were similar between the two cohorts. Again, this is replicated in the literature [7,8].

### 4.4. Potential Benefits of HFRT

In the setting of comparable outcomes, HFRT presents additional benefits in comparison to BID schedules. Primarily, HFRT is easier to administer from a logistical standpoint for both patients and treating professionals [16,30,36]. Patients treated on BID schedules require at least a 6 h period in between fractions, which may take a physical, mental, and financial toll. The magnitude of effect of patient inconvenience in the selection of dose fractionation has yet to be formally evaluated from a patient standpoint, although previous survey data suggest that it is a primary consideration for radiation oncologists [6,37]. Secondly, the higher rates of toxicity observed in the Turrisi trial have introduced a persistent hesitancy in the universal adoption of BID treatment. A survey of experienced European radiation oncologists revealed that most institutions preferred once-daily treatment, albeit of conventional fractionation, for unfit patients [16,36]. However, toxicity may be similar between BID and HFRT, as suggested by the current analysis and previous studies, particularly in the advent of more precise radiotherapy technologies such as 4D-CT simulation and IMRT [7,8]. 

Furthermore, HFRT circumvents some of the concerns regarding accelerated repopulation that protracted radiotherapy schedules, such as 60–66 Gy in 30–33 fractions, may present [38,39]. It is well established that early administration of radiation concurrently with chemotherapy improves outcomes. However, the total duration of radiotherapy, defined as SER by De Ruysscher et al., showed that each week extension in SER was associated with a nearly 2% decrease in 5-year OS in a meta-analysis [14]. Their subsequent individual patient meta-analysis also showed that “shorter or earlier” thoracic radiotherapy administration in comparison to “longer or later” was associated with a nearly 8% improvement in 5-year OS when comparing patients with similar chemotherapy adherence [40]. 

Nevertheless, despite its potential advantages, there is hesitancy in the recommendation of HFRT as a standard radiotherapy schedule in thoracic CRT for LS-SCLC patients. The recent ASTRO guidelines recommend 45 Gy/30 BID as a standard treatment, and that 60–70 Gy in conventional fractionation is an acceptable alternative. HFRT was not routinely recommended owing to the limited evidence for its equivalence [1]. Similarly, despite the opportunity for minimizing patient-viral exposure that HFRT presents, the COVID-19 ASTRO-ESTRO consensus guideline only recommends HFRT in late-phase pandemic conditions, when radiotherapy resources are limited [41]. 

### 4.5. Strengths and Limitations

The strengths of our study include our collection of a comprehensive set of covariates thought to potentially confound the effect of radiotherapy schedule on the outcomes of interest. We adjusted for these confounders using propensity score weighting and created a cohort with nearly perfect balance on covariates. E-value calculations demonstrate robust estimates of OS, LRR, and toxicity outcomes, in which only unobserved confounders of substantial magnitude could alter the observed associations. 

Our study is inherently limited by its retrospective nature in terms of data quality, confounding, and generalizability. One covariate that was unavailable within our dataset was the extent of the target volume; however, our analyses did include AJCC stage, which is also representative of disease burden. We addressed observed confounding with adjustment using overlap weighting to balance measured covariates. Despite the E-value estimates, the possibility of unobserved confounding persists after overlap weighting and cannot be completely accounted for. Regarding generalizability, the effects estimated using the overlap weights generalize only to patients in the overlap population rather than to the LS-SCLC population as a whole. However, the impact of using overlap weights here on generalizability is minimal, as the distribution of patient characteristics in the overlap-weighted sample did not differ greatly from that in the original sample. In regards to missing covariate values, we utilized multiple imputation, which we recognize does not represent true data values and assumes that missing values do not depend on unobserved factors. However, the amount of missing data was small, and previous evidence suggests that it can be valid for covariates such as stage [42]. Lastly, we do caution the interpretation of the results, in that the lack of treatment effect between the two radiotherapy groups may be in part attributed to a lack in statistical power to determine a difference, rather than there truly being no difference. Nevertheless, previous literature supports the observed equivalence of BID and HFRT schedules [7,8,43]. 

## 5. Conclusions

Concurrent CRT with HFRT compared to BID radiotherapy did not show a difference in OS, LRR, or ≥grade 3 toxicities in our propensity score adjusted LS-SCLC cohort. While BID treatment remains the gold standard, the use of HFRT as a suitable treatment schedule warrants consideration when clinically appropriate due to the shorter treatment duration and logistical benefits. Prospective study to establish the effectiveness or non-inferiority of HFRT in comparison to other radiotherapy schedules for LS-SCLC is merited.

## Figures and Tables

**Figure 1 cancers-13-02895-f001:**
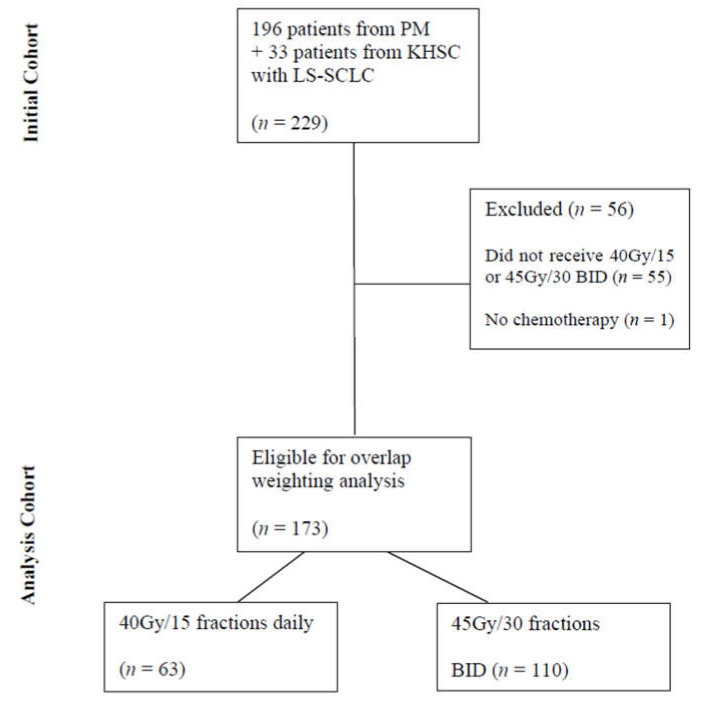
Flow Diagram Detailing Patient Selection. Abbreviations: PM—Princess Margaret Hospital, KHSC—Kingston Health Sciences Centre, LS-SCLC—limited-stage small-cell lung cancer, BID—twice daily.

**Figure 2 cancers-13-02895-f002:**
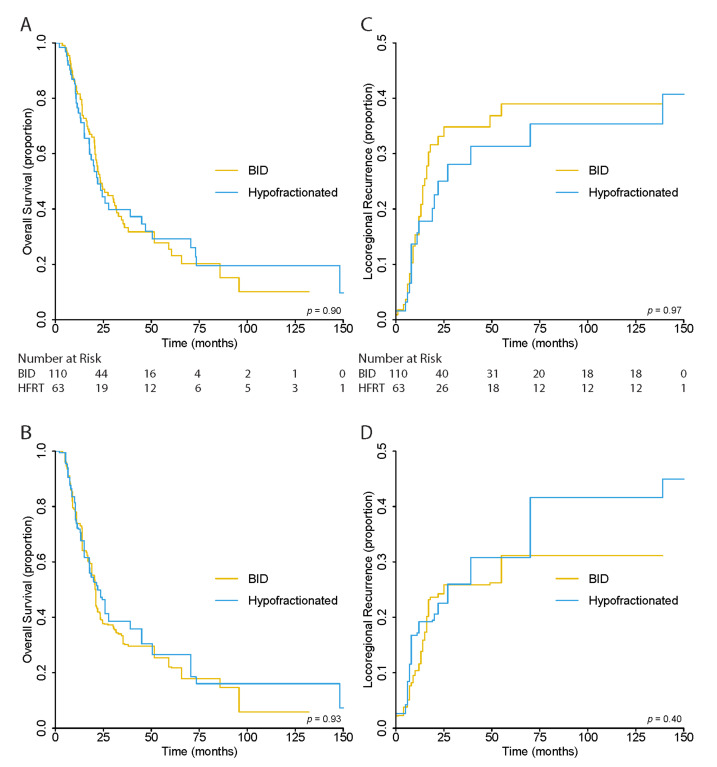
(**A**) Unweighted and (**B**) Overlap-Weighted Kaplan–Meier Estimates for Overall Survival and (**C**) Unweighted and (**D**) Overlap–Weighted Cumulative Incidence Functions for Locoregional Recurrence Risk.

**Table 1 cancers-13-02895-t001:** Baseline Characteristics of Original and Weighted Cohorts.

Variables	All (*n* = 173)	45 Gy/30 Fractions BID (*n* = 110)	40 Gy/15 Fractions (*n* = 63)	*p*-Value ***	Pre-Weighting SMD *	Post-Weighting SMD *
Age (mean, SD)	66.7 (9.7)	65.7 (8.9)	68.5 (10.8)	0.12	0.33	0.001
Gender (*n*, %)				0.59	0.04	0
Male	98 (57)	64 (58)	34 (54)			
Female	75 (43)	46 (42)	29 (46)			
Surgery (*n*, %)				0.03	0.09	0
Yes	12 (7)	4 (4)	8 (13)			
No	161 (93)	106 (96)	55 (87)			
ECOG (*n*, %)				0.80	0.65	0
0	47 (27)	29 (26)	18 (29)			
1	84 (49)	56 (51)	28 (44)			
2	33 (19)	19 (17)	14 (22)			
3	9 (5)	6 (6)	3 (5)			
Stage (*n*, %)				0.13	0.13	0
IA	16 (9)	5 (4)	11 (17)			
IB	5 (3)	1 (2)	3 (5)			
IIA	5 (3)	3 (3)	2 (3)			
IIB	11 (6)	8 (8)	3 (5)			
IIIA	51 (29)	35 (32)	16 (25)			
IIIB	58 (34)	39 (36)	19 (30)			
IIIC	27 (16)	18 (16)	9 (14)			
Year of Treatment (median, IQR)	2013 (2009–2016)	2014 (2010–2016)	2011 (2008–2016)	0.02	0.17	0
Smoking Status (*n*, %)				1.0	0.005	0
Never/Not Documented	6 (3)	4 (4)	2 (3)			
Former	112 (65)	71 (64)	41 (65)			
Current	55 (32)	35 (32)	20 (31)			
Pack Years (mean, SD)	43.34 (21.52)	41.91 (21.66)	45.84 (21.22)	0.16	0.18	0
Paraneoplastic Syndrome (*n*, %)				0.89	0.007	0.003
Yes	20 (12)	13 (12)	7 (11)			
No	153 (88)	97 (88)	56 (89)			
mCCI Score (*n*, %)				0.79	0.05	0
0	114 (66)	74 (67)	40 (64)			
1	47 (27)	28 (25)	19 (30)			
2+	12 (7)	8 (7)	4 (6)			
PET-CT scan (*n*, %)				<0.01	0.25	0
Yes	96 (55)	71 (65)	25 (40)			
No	77 (45)	39 (35)	38 (60)			
Pre-treatment Brain Imaging (*n*, %)				0.35	0.06	0
None	1 (1)	0 (0)	1 (1)			
CT	45 (26)	27 (25)	18 (29)			
MRI	127 (73)	83 (75)	44 (70)			
4D-CT Utilization (*n*, %)				0.48	0.01	0
Yes	158 (91)	101 (92)	57 (90)			
No	15 (9)	9 (8)	6 (10)			
Treatment Technique (*n*, %)				0.01	0.20	0
3D-CRT	22 (13)	9 (8)	13 (21)			
IMRT	135 (76)	92 (84)	40 (63)			
VMAT	19 (11)	9 (8)	10 (16)			
IGRT Utilization (*n*, %)				<0.01	0.24	0
Non-IGRT	39 (23)	16 (15)	23 (37)			
CBCT	134 (77)	94 (85)	40 (63)			
** PCI Utilization (*n*, %)				0.29		
Yes	121 (70)	80 (73)	41 (65)			
No	52 (30)	30 (27)	22 (35)			
Chemotherapy (*n*, %)				<0.01	0.15	0
Concurrent	154 (89)	104 (95)	50 (79)			
Sequential	19 (11)	6 (5)	13 (21)			
Cycles of Chemotherapy (*n*, %)				0.12	0.36	0.03
0	1 (1)	0 (0)	1 (2)			
1	3 (2)	1 (1)	2 (3)			
2	9 (5)	6 (5)	3 (5)			
3	9 (5)	3 (3)	6 (10)			
4	63 (36)	37 (34)	26 (41)			
5	18 (10)	13 (12)	5 (8)			
6	70 (40)	50 (45)	20 (32)			
Chemotherapy to RT Time in Days (mean, SD)	56.05 (36.80)	50.71 (32.37)	65.38 (42.14)	0.05	0.31	0

Abbreviations: BID—twice daily, ECOG—Eastern Cooperative Oncology Group, mCCI—modified Charlson Comorbidity Index, IQR—interquartile range, PET—positron emission tomography, 4D-CT—4-dimensional computed tomography, VMAT—volumetric modulated arc therapy, IMRT—intensity modulated radiotherapy, 3D-CRT—3-dimensional conformal radiotherapy, IGRT—image guided radiotherapy, KV-KV—orthogonal kV films, CBCT—cone beam CT, PCI—prophylactic cranial irradiation, SMD—standardized mean difference; * Note that for non-continuous covariates, the displayed SMD is the largest applied across all categorical levels. ** PCI was not included in covariate balancing since its utilization is determined after the treatment variable, thoracic radiotherapy, has been administered; *** *p*-values determined from Mann–Whitney U test and Fisher’s exact test for continuous and categorical variables respectively.

**Table 2 cancers-13-02895-t002:** Univariable and Multivariable Regression Models for Radiotherapy Schedule Effect on Outcomes in Weighted and Unweighted Populations (HFRT versus BID [reference]).

Outcome	Unweighted				Overlap Weighted			
	Univariable		Multivariable		Univariable		Multivariable	
	HR/OR (95% CI)	*p*-Value	HR/OR (95% CI)	*p*-Value	HR/OR (95% CI)	*p*-Value	HR/OR (95% CI)	*p*-Value
OS	0.84 (0.48–1.51)	0.57	0.72 (0.36–1.40)	0.32	1.16 (0.58–2.29)	0.67	1.67 (0.70–3.95)	0.25
LRR risk	0.86 (0.49–1.53)	0.62	1.33 (0.64–2.75)	0.44	1.29 (0.61–2.72)	0.51	1.48 (0.62–3.54)	0.38
Thoracic Response	3.89 (1.12–13.57)	0.03	1.00 (0.22–4.64)	1.00	1.00 (0.01–4.73)	1.00	0.23 (0.02–2.23)	0.21

Abbreviations: HFRT—hypofractionated radiotherapy, BID—twice-daily radiotherapy, CI—confidence interval, OS—overall survival, LRR—locoregional recurrence, OR—odds ratio, HR—hazard ratio.

**Table 3 cancers-13-02895-t003:** Univariable and Multivariable Regression Models for Radiotherapy Schedule Effect on Grade 3+ Toxicity in Weighted and Unweighted Populations (HFRT versus BID [reference]).

Toxicity	Unweighted				Overlap Weighted			
	Univariable		Multivariable		Univariable		Multivariable	
	OR (95% CI)	*p*-Value	OR (95% CI)	*p*-Value	OR (95% CI)	*p*-Value	OR (95% CI)	*p*-Value
Any Toxicity	1.29 (0.14–0.36)	0.52	1.31 (0.59–2.89)	0.51	1.62 (0.58–4.53)	0.36	1.67 (0.59–4.72)	0.33
Pulmonary	1.29 (0.39–4.27)	0.68	1.16 (0.31–4.30)	0.82	1.06 (0.25–4.50)	0.93	1.14 (0.32–4.10)	0.84
Esophageal	1.15 (0.47–2.84)	0.76	1.19 (0.47–3.02)	0.72	1.38 (0.38–5.00)	0.63	1.41 (0.36–5.51)	0.62

Abbreviations: HFRT—hypofractionated radiotherapy, BID—twice daily radiotherapy, CI—confidence interval, OR—odds ratio.

**Table 4 cancers-13-02895-t004:** Studies Including Patients with Limited-Stage Small-Cell Lung Cancer Treated with Hypofractionated Radiotherapy.

Author	Year	HFRT Schedule (Gy/fx)	*n*	Overall Survival (%)		≥Grade 3 Esophageal Toxicity (%)	≥Grade 3 Lung Toxicity (%)
				2-Year	5-Year		
Murray et al. [10] *	1993	40/15	155 ^#^	40	20	43.6	3.2
Videtic et al. [32]	2003	40/15	122	27	9	-	-
Bettington et al. [8]	2013	40/15	38	-	20	-	-
Socha et al. [33]	2015	42/15	100	52	31	0	0
Grønberg et al. [7]	2016	45/15	84	42	-	31	6
Turgeon et al. [31] *	2017	40/15	68	53	35	9	1
Zhang et al. [34]	2017	55/20	69	62	-	12	10
Zayed et al. [9]	2020	40–45/15–20	56	-	26	6+	3+
Present study	2020	40/15	63	47	24	14	8

Abbreviations: HFRT—hypofractionated, Gy—gray, fx—fractions, OS—overall survival, LRR—locoregional recurrence, GI—gastrointestinal; * Randomized trials; ^#^ Only the early concurrent therapy arm is included.

## Data Availability

The datasets used in the current study are available from the corresponding author upon request.

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
