# Peer review of "A Comparison of Hypofractionated and Twice-Daily Thoracic Irradiation in Limited-Stage Small-Cell Lung Cancer: An Overlap-Weighted Analysis"

_cancers, 2021, doi:10.3390/cancers13122895_

Round 1
Reviewer 1 Report
Overall this paper well presented and structured, and addresses an interesting and relevant question. Limitations due to retrospective study, but these limitations are recognised by the authors. With respect to the results, some of language and specific description around the analysis eg. the Kolmogorov-Smirnov (KS) statistic is somewhat complicated and likely unnecessary. The authors could consider removing/refining some of this.
Author Response
Thank you for your review. After discussion with our statistical team, we have omitted the sentence describing the KS statistic. It is an adjunct measure of balance, which is primarily described by the standardized mean difference (SMD). We feel that the SMD alone is sufficient to indicate good balance.
Reviewer 2 Report
The purpose of this study is relevant. We need the evidence that accelerated hypofractionated RT is a valuable option in management of LS-SCLC. Unfortunately, this convenient and commonly used schema was not tested in any large randomized trials. We have some observational studies that compare hypofractionated RT with conventionally fractionated RT that showed no inferiority of hypofractionation (cited in the manuscript: Zayed IJROBP 2020, but else not mentionned Videtic IJROBP 2003, Socha J BUON 2015, Zhang Radiat Oncol 2017). However, the retrospective comparisons of hypofractionation with hyperfractionation are more limited. To my knowledge, only cited by the authors Bettington CS. Thus a contribution of the authors is relevant.
I have some minor comments that should be addressed:
- Lines 48-51: Ref.4 is inadequately cited: because the CONVERT trial did not establish a superior survival with higher toxicity of hyperfractionation over conventional fractionation
- Please, refer to other observational studies that were in line with the findings from this study.
- One of the variables that impact outcome of RT is the volume of the disease. It is possible to include GTV as the variable in the statistical analysis; if not, this should be acknowledged.
Author Response
Thank you for your comments, they are well received. To address the comments:
- We have edited the introduction to clarify the detail that CONVERT did not demonstrate superiority of 66Gy/33 daily compared to 45BID, as opposed generalizing that BID had improved survival compared to daily treatment. Further elaboration of these two seminal trials are detailed within the discussion.
- We have added Videtic et al. and Socha et al. as references within the discussion in section 4.3. We also added them to Table 4. Zhang et al. has previously been included in Table 4.
- Unfortunately, GTV/disease volume was not recorded. We added this to our list of limitations in section 4.5 of discussion. We did however adjust for stage, which is also representative of disease burden.
Reviewer 3 Report
This is a well-designed and well-executed retrospective comparison of two common fractionation schedules in Canada for the definitive treatment of LS-SCLC. The statistics are sophisticated and well-described, and the results are reported clearly. The writing is excellent. I do not see any major methodological issues.
The study is limited by 1) the relatively small sample size, particularly in comparison to published randomized studies of various dose/fractionation schemes with similar patient numbers; 2) the limited use of hypofractionated RT for LS-SCLC outside of Canada (at least to my knowledge), which limits the impact of this work; 3) recent randomized evidence (Gronberg et al. Lancet Oncol 2021) which suggests a substantial survival benefit to dose-escalated hyperfractionated RT, further limiting the impact of the present work.
Despite these issues, in the future hypofractionated RT may continue to be commonly utilized in Canada and the present data are useful in supporting that practice.
I would request that the authors revise their Discussion to include the recent findings from Gronberg et al (citation below) and explore how these data affect the relevance of their work.
Grønberg, Bjørn Henning, Kristin Toftaker Killingberg, Øystein Fløtten, Odd Terje Brustugun, Kjersti Hornslien, Tesfaye Madebo, Seppo Wang Langer, et al. 2021. “High-Dose versus Standard-Dose Twice-Daily Thoracic Radiotherapy for Patients with Limited Stage Small-Cell Lung Cancer: An Open-Label, Randomised, Phase 2 Trial.” The Lancet Oncology 22 (3): 321–31.
Author Response
Thank you for your comments, and for bringing this important trial to our attention.
We would like to note however, that this trial does not directly investigate the current study question (HFRT vs BID), but instead suggests that dose escalated BID treatment is better than standard 45Gy/30 BID. Additionally, some caveats of the trial includes the observed OS benefit without an associated PFS benefit, which calls into question the mechanism of this survival increase.
Nevertheless, we have included the recent trial by Gronberg et al. in section 4.1 of the discussion in order to provide a comprehensive overview of the literature.